# First Results on Zinc Oxide Thick Film Deposition by Inverted Magnetron Sputtering for Cyclotron Solid Targets Production

**DOI:** 10.3390/ma16103810

**Published:** 2023-05-18

**Authors:** Alisa Kotliarenko, Oscar Azzolini, Sara Cisternino, Mourad El Idrissi, Juan Esposito, Giorgio Keppel, Cristian Pira, Angelo Taibi

**Affiliations:** 1Legnaro National Laboratories, Italian National Institute for Nuclear Physics (LNL-INFN), 35020 Legnaro, Italy; oscar.azzolini@lnl.infn.it (O.A.); sara.cisternino@lnl.infn.it (S.C.); mourad.el.idrissi@lnl.infn.it (M.E.I.); juan.esposito@lnl.infn.it (J.E.); giorgio.keppel@lnl.infn.it (G.K.); cristian.pira@lnl.infn.it (C.P.); 2Department of Physics and Earth Science, University of Ferrara, 44122 Ferrara, Italy; taibi@fe.infn.it; 3Department of Industrial Engineering, University of Padua, 35131 Padua, Italy

**Keywords:** radioisotope production, magnetron sputtering, ultra-thick deposition, material losses, cyclotron solid targets, ZnO

## Abstract

The magnetron sputtering technique has been investigated in recent years with ever-growing interest as a verifiable solid target manufacturing technology aimed at the production of medical radionuclides by using low-energy cyclotron accelerators. However, the possible loss of high-cost materials prevents access to work with isotopically enriched metals. The need for expensive materials for the supply of the growing demand for theranostic radionuclides makes the material-saving approach and recovery essential for the radiopharmaceutical field. To overcome the main magnetron sputtering drawback, an alternative configuration is proposed. In this work, an inverted magnetron prototype for the deposition of tens of μm film onto different substrates is developed. Such configuration for solid target manufacturing has been proposed for the first time. Two ZnO depositions (20–30 μm) onto Nb backing were carried out and analysed by SEM (Scanning Electron Microscopy) and XRD (X-ray Diffractogram). Their thermomechanical stability under the proton beam of a medical cyclotron was tested as well. A possible improvement of the prototype and the perspective of its utilisation were discussed.

## 1. Introduction

Magnetron sputtering (MS) is a well-known technique for thin film deposition, which is adopted in different industrial and scientific areas. In the medical field, MS is widely used for protective and antibacterial coating deposition for metal implants or surgical instruments [1,2]. Some works also report its potential use in the nuclear medicine field as a corrosion-resistant coating for liquid cyclotron targets [3,4]. In recent years, the MS has been also investigated, referring to the solid target manufacturing for radiopharmaceutical production [5,6,7,8]. Among the advantages of the MS technique, thickness control and high adhesion between the film and substrate may be outlined. The possibility to manufacture solid targets by using MS for different radionuclides production, such as ${}^{99m}Tc$ and ${}^{89}Zr$, has already been discussed [5,6].

The interest in solid target manufacturing has been consistently growing over recent years. Such targets can be used for the production of conventional and non-conventional medical radionuclides. The design of the solid targets can be different depending on the desirable characteristic and its specific application. Targets can be self-supported or attached to a backing of different materials. The targets used for the production of conventional or emerging medical radionuclides are usually made of isotopically enriched material (metal or compound) to obtain high radionuclidic purity. Such materials are consistently expensive [9]. However, some radioisotopes can be produced by monoisotopic compounds, as in the case of ${}^{89}Zr$ produced by *Y* solid targets. In any case, the configuration of the solid target has always met the main requirements: chemical purity and defined thickness, thermo-mechanical stability under the beam as well as ease of handling in the hot cell where the radiochemistry process of dissolution occurs. Moreover, not only the target properties but also the economic aspect of such products must be considered [6,8,10].

In the previous work [7], an estimation of the material losses during the MS process using a standard planar magnetron configuration was carried out. The results show a very low process efficiency from the material point of view—more than 50% is usually lost in the vacuum chamber. Therefore, it is proposed to use a recovery shield, which allows for a significant amount of material to be collected and then recovered. Even in such a case, losses are at the level of 5%, and the material recovery will require additional time and money investment. Unfortunately, due to low material efficiency sputtering technique does not allow to work with an isotopically-enriched high-cost material.

The idea of this work is to find a way to decrease losses of the initial material and make the sputtering technique attractive also in the case of expensive isotopically-enriched materials. This work is a part of the numerous activities presented by the LARAMED group [11] of INFN-LNL and aimed at the deep evaluation of the MS technique applied to solid target manufacturing for medical radioisotope production, although focused on the material-saving approach.

In this work, a different approach has been employed for the first time in the production of a solid target by utilising an inverted magnetron (IM) configuration. The IM has a cylindrical shape and may also be called a hollow cathode. Different cylindrical IM configurations have been commercialised, indeed, the idea of such configuration is not new and has been studied for decades [12,13]. Some work reports different IM sources for the deposition of thin metal films and compounds [14,15].

The potential of the MS technique implemented in this work can be exploited to expand the number of radionuclides available for diagnostics ${}^{99m}Tc$, ${}^{89}Zr$, ${}^{52}Mn$, ${}^{64}Cu$, and for the therapy. With this regard, it is important to emphasise the versatility of the theranostic (therapy + diagnostics) radionuclide ${}^{67}Cu$, under the spotlight of the scientific community [16,17,18]. Although the nuclear physics reactions to optimise the ${}^{67}Cu$ production have been extensively studied [17], its production in a sufficient amount for pre-clinical as well as clinical tests is limited by the lack of a suitable target that can withstand the high proton beam current. For this reason, the development of innovative target manufacturing technologies is fundamental. The starting materials for the production of ${}^{67}Cu$ are the isotopically enriched ${}^{68,70}Zn$. Since the melting point of Zn is relatively low ($T_{m}$ = 420 °C), and it can be damaged under the high current proton beam, ZnO ($T_{m}$ = 1975 °C) has been considered as well.

In this study, thick ZnO film by reactive sputtering deposition starting with metallic Zn cathode material was investigated. Zinc oxide has unique properties [19] and can be used in different applications, such as a transparent conductive oxide (TCO) [20,21], piezoelectric devices, chemical sensors [19,22], biosensors, and many others. Many works report ZnO deposition by using a sputtering technique with planar sputtering source, however, emphasising radio frequency (RF) over the direct current (DC) deposition [19,20,21,22,23,24,25,26,27]. The particular interest for this material is also rising since it is considered a possible cheaper substitution of commonly used ITO (indium tin oxides) in microelectronic devices [20,28]. However, in all mentioned studies where ZnO is deposited, the thickness ranges from 50 to 2000 nm. For the solid targets manufacturing, the thickness typically starts from tens of μm for nuclear cross-section analysis and can reach up to tens of mm for maximizing radionuclide production [9].

Reporting a classification of the various methods that can be used for solid target manufacturing is not easy. In the work of [9], the authors suggested dividing them into three large groups: mechanical reshaping with power processing and physical and chemical methods. Each method has its own benefits and limitations. For example, standard powder pressing is a relatively simple and quick method, especially whether the material of interest is available in powder form. The losses are minimal, and the target can be produced with a very small amount of material. The limitations, in the case of radioisotope production, are the mechanical properties of the achieved targets. In the case where the target may endure a high beam power and for an extended duration, advanced sintering methods such as spark plasma sintering may be recommended [10,29].

In this work, an inverted magnetron source prototype was designed and proposed as an alternative solid target manufacturing technique for the production of medical radionuclides by irradiation through medical cyclotrons. It is shown that achieving the deposition of thick ZnO films is really feasible. In particular, the preliminary study on the deposition of the cheapest ${}^{nat}ZnO$ (20–30 μm) onto Nb substrates has led to the manufacturing of the first ${}^{nat}ZnO$ targets that have been tested under the proton beam of a medical cyclotron.

In this regard, it is important to mention that radiopharmaceutical production is a complex chain of different processes, including target manufacturing, cyclotron irradiation, chemical dissolution, purification processes, quality control, and labelling [8]. The irradiation test discussed in this work shows preliminary results of deposits’ termo-mechanical properties. Further steps, such as the dissolution of the ZnO deposits, were not undertaken, as the required elevated thicknesses had not yet been achieved.

It should be noted that the term ‘target’ is commonly used in the radiopharmaceutical field as a starting material for radionuclide production. In the sputtering field, it refers to a material to be sputtered onto a substrate, also known as a cathode or sputtering material. To avoid confusion, this article will use ‘target’ in reference to the radiopharmaceutical field and ‘cathode’ in reference to the sputtering material.

## 2. Materials and Methods

### 2.1. Brief Description of Inverted Magnetron Source

The cylindrical (both post and hallow) magnetron is a device that combines simple geometry with high performance as a sputtering source [12]. Figure 1 illustrates the scheme of the sputtering system by using an inverted magnetron source and a dedicated sample holder with a mask.

The deposition is carried out in a vacuum, with the presence of inert gas. This gas gets ionised near the cylindrical cathode (made of the material to deposit) and, due to the application of a magnetic field and the presence of an electric field, creates a high-density plasma. Gas ions bombard the negatively charged cathode and extract atoms from it. The atoms are then ejected from the cathode and move in all directions inside the vacuum chamber, thus, depositing on the substrate.

Typically, the electric potential difference is created by powering with a negative charge the cathode material and leaving the sample holder grounded, floated, or biased. However, in our case, such voltage was created by giving the sample holder a positive charge, leaving the cathode and the chamber at the ground potential. This approach was chosen since it allows simpler source assembling and operating.

As a first step, a simplified IM prototype was designed and developed with the aim of evaluating the possibility of using this configuration for solid target manufacturing. The prototype has the following dimensions: 106 mm height, 89 mm external, and 59 mm internal diameter.

Another important aspect to be considered is the confinement of the plasma inside the closed cathode volume, otherwise, the stainless steel (SS) components of the vacuum chamber would also be sputtered and deposited onto the substrate. To fix this issue additional bottom and top plates (see Figure 1 and Figure 2b), made of cathode material, were applied. The gas circulation in the closed cathode environment is ensured by adding holes in the top shield. As an alternative upgrade, both shields can be produced in quartz or glass materials.

The static magnetic field is created by an external 3-coil system resulting in a 250 G magnetic field at the central position. The magnetic field was evaluated by LakeShore 421 Gaussmeter. A dedicated power supply, Sorensen DCS80-15E, with a maximum of 80 V voltage, was used to power the coils.

A dedicated sample holder was realised to be then used in vertical or horizontal positions, thus, to be parallel or perpendicular to the cathode walls. The sample holder, made in SS, was sized to be compatible with the prototype, and it can be used to accommodate substrates of different diameters (Ø24 and Ø13 mm) and thicknesses (1.2 and 0.3 mm) through the use of the different masks. The mask’s apertures are chamfered to promote a uniform deposition above the uncovered part of the substrate, as it is a mandatory requirement for targets in radioisotope production. Figure 2a also shows how the sample holder is assembled with the mask inside the IM. In this case, the mask is dedicated to Nb coin (or disc) (diameter 24 mm, thickness 1.2 mm) used in this work as a backing material for the target.

Different substrates materials were used in this study to characterise the achieved deposits: Si wafers—for cross-section SEM analysis and quartz samples (10 × 10 mm^2^) for profilometer and XRD analysis.

All SS components were properly cleaned by applying the following procedures:ultrasound bath with distilled water with the addition of 5 mL of Rodaclean^®^ (NGL Cleaning Technology SA, Nyon, Switzerland) soap for 20 min at 40 °C;ultrasound bath with distilled water for 20 min;rinsing with ethanol 96% and drying with the $N_{2}$ gas gun (oil-free compression system).

### 2.2. Sputtering System

The sputtering process was held in the SS cylindrical vacuum chamber with a volume of 5.8 dm^3^. The base vacuum level was created by a primary and turbo molecular pump (the Edwards nXDS6i of 100 L/min and Pfeiffer turbo molecular pump of 60 L/s, respectively) with a base pressure always lower than 6 × 10^−4^ Pa without any additional baking procedure.

The deposition system is connected to the two working gases: Ar (99.9999% purity), used as a sputtering inert gas, and $O_{2}$ (99.9999%), used as the reactive gas for oxide creation. The sputtering process was performed using a power supply DC Pinnacle (20 kW) from Advanced Energy with a 3-phase connection, which allows it to work with positive potential.

For the preliminary system adjustment, the copper (99.9%) cathode was used as a sputtering material. The first cathode was made of 0.5 mm foil by manual bending, which then was replaced with a cathode prepared at LNL mechanical workshop. The procedure includes machine bending, welding, and post-polishing with abrasive paper. The zinc cathode for forthcoming tests was produced from Zn foil (99.95%) 1 mm thick. The cleaning procedure followed for the Cu and Zn cathodes was the same as that used for the standard cleaning protocol adopted, but with a different detergent agent, the GP 17.40 SUP soap (NGL Cleaning Technology SA, Nyon, Switzerland).

### 2.3. Characterisations

An evaluation of the produced targets was carried out through a set of analyses. The density calculations were performed using information on the weight and thickness of obtained films. The weight measurements were performed with electronic precision balances (Sartorius MCA225S-2S00-I Cubis^®^ II Semi Micro Balance, 220 g × 0.01 mg). The thickness was measured by a linear contact profilometer (Veeco Dektak 8). The growing behaviour and thickness of the films were accurately investigated by scanning electron microscopy (COXEM, CX-200plus). Additionally, X-ray Diffractograms were collected (X’Pert Powder, Philips) and analysed with dedicated HighScorePlus software [30]. The diffractogram was acquired by using Cu-Kα X-ray λ = 1.54 Å as the radiation source at 40 kV and 40 mA.

### 2.4. Cyclotron Irradiation

The irradiation tests of the targets at increasing current are commonly performed in the early stages of target manufacturing optimisation. Thus, in this work, the thermo-mechanical stability of the first ZnO targets manufactured with the new IM set-up was tested using a medical cyclotron following the approach described in [29].

The proton irradiation was carried out on a variable energy (14–19 MeV) TR19 cyclotron (ACSI, Richmond, BC, Canada) equipped with a high-current ion source up to 300 μA. The target station (ACSI) is oriented at 90°, and it is directly connected to the cyclotron target selector. The target coin is cooled down by flushing helium gas onto the front side of the target and by water from the backside. The cyclotron target holder can hold a coin with a maximum thickness of 2 mm and a diameter of 24 mm.

After the irradiation, the targets were visually inspected to assess their integrity as a common practice in target manufacturing [5,6,10].

## 3. Results

### 3.1. Deposition Parameters and Setup Optimisation

To find out the working condition of the IM prototype, more than 30 tests were performed using a Cu cylindrical cathode and sample holder of different orientations (horizontal and vertical). As a result, an optimal configuration with a horizontally placed sample holder was found, and a 60 μm Cu deposition was carried out onto Nb backing with a deposition rate of 0.4 μm/min and current of 1 A. Higher currents were not applied to avoid overheating problems since a cooling system adapted to the sample holder was not provided.

The coil position and the symmetry of all components (cathode, sample holder, samples) play a key role in the stability of the process during the deposition run. For this reason, an additional component, that is, a Cu central ring, was used to place the magnetron in the centre of the vacuum chamber.

The first batch of tests with Zn cathode was performed to find out the best working parameters to manufacture a ZnO compound and to maximise the deposition rate. The working pressure was fixed at 2.9 Pa with the Ar:$O_{2}$ = 2:1 gas composition. Table 1 reports the parameters for ZnO deposition: exp 8 was used for the cross-section, and Targets 1 and 2 were deposited as the first solid target prototypes. All the tests were performed with 0.7 A. The limitation of the deposition time was mostly due to the flakes formation, which resulted in the arcs and instability of the process.

As a result, two ZnO films were deposited on the Nb disks with a deposition rate of 150–200 nm/min. The density calculation was conducted using the weigh difference method, shown in Table 1. The relative density is found to be around 70% of the bulk ZnO (5.67 g/cm^3^) [31]. To understand the nature of these values, morphological analyses were performed using SEM.

The effect of the temperature was not studied in this work, however, the overheating of the sample holder and, thus, samples were evident immediately after the current was switched off by the visual inspection through the viewport. Some considerations about the temperatures will be described in the Discussion Section 4.

### 3.2. Morphology, Growing Behaviour, and Deposition Rate

The SEM images presented in Figure 3 show the morphology of the deposits of Target 1 and 2. The presence of massive macro-particles and flakes is apparent (sizes around 100 μm), which, sometimes, remained embedded into the film (see Figure 3a,c) or was removed, thus, leaving craters-like defects. As seen in Figure 3b, the detachment happened during the ongoing deposition; hence the crater was redeposited. Both these phenomena can make it difficult to evaluate the thickness with the standard contact profilometer introducing an unpredictable error to the measurements.

To study the growth behaviours of the obtained films, SEM cross-section analysis was performed on the deposition carried out on Si substrate (exp 8). The SEM micrography is shown in Figure 4. Two distinct growth patterns are observable. The initial 2–2.5 μm deposition exhibits uniform columnar growth, followed by a transition to the formation of larger crystals.

### 3.3. Compound Evaluation

The XRD analysis was performed for ZnO film on Nb and quartz samples to confirm the formation of the stoichiometric ZnO. The results are shown in Figure 5 and were compared to the diffractogram of ZnO powder.

The evaluation of the obtained diffractograms in HighScorePlus software confirms the formation of the ZnO film without the presence of metallic Zn. However, in the diffractogram of ZnO onto Nb, the Nb peaks are visible due to the high penetration depth of the X-rays in the chosen instrument configuration. The diffractogram obtained from the ZnO on the quartz sample also supports the obtained results.

It is important to note that the sputtering processes typically exhibit preferential film growth. In our case, the higher peak is attributed to the low energy needed for the formation of the (002) plane, which could be assigned to the hexagonal ZnO würtzite phase with the c-axis perpendicular to the substrate surface, as reported in several works [27,32,33].

Crystallite sizes were calculated from the Scherrer formula: $D=\frac{K\lambda }{\beta \cos\theta }$, where *K* is the shape factor of the average crystallite, λ is the X-ray wavelength (λ = 0.15406 nm for *Cu* target), β is the full width at half-maximum in radians, θ is the Bragg angle, and *D* is the mean crystallite dimension normal to diffracting planes [34]. Based on the calculation, the crystalline size of ZnO on the Nb coin appears to be ∼30 nm.

### 3.4. Cyclotron Irradiation Tests

As a final step, the produced ZnO targets were tested under the cyclotron beam. The thermo-mechanical stability of the sputtered targets was evaluated with two irradiation runs (energy of 19 MeV) at 10 and 20 μA for 5 min. Visual control of the targets after each irradiation was carried out. The photos of the targets before and after the irradiation are shown in Figure 6.

The sputtered targets showed no visible signs of delamination or cracking following the irradiation tests. These outcomes can be explained by high adhesion performance, facilitating efficient heat exchange during irradiation to dissipate the heat power proportional to the current deposited on the target.

## 4. Discussion

### 4.1. IM Sputtering Process

This work shows an alternative path of the material-saving approach of sputtering deposition dedicated to solid targets manufacturing aimed at medically-relevant radioisotope production. The described IM configuration uses the principle of sputtering, which was described by Thornton in 1977 [12]. The main difference of the IM prototype used in this work with respect to the standard IM deposition process was the positive potential connection to the sample holder (see Figure 1), instead of the more common negative cathode connection [14]. This aspect might have some beneficial effects on the deposition process and deposit growth. Such configuration prevents an overheating of the Zn cathode, which has a low melting temperature and probably causes some overheating of the sample holder, where ZnO was sputtered.

Temperature data were not accessible, but accurate structural analysis can provide an idea of the reached temperatures. The Structural Zone Diagram (SZD) of Thornton adapted to $CuO_{2}$ deposition, presented in the work [35], can be used to analyse ZnO performance. The structure of the surface and cross-section of the film (see Figure 4) can be compared to our ZnO to Zone 2, where $T_{h}=0.5–0.6$. Additionally, the SS sample holder turned to a red hot colour immediately after the power was switched off, indicating a temperature range of approximately 500–800 °C. While these values are not a precise indication of the temperature, they provide an idea of the range that correlates with the SZD. This suggests that our deposition process was facilitated at a high temperature of the substrate, which may have played a role in adhesion, deposition rate, and residual stress reduction.

The reactive magnetron sputtering process is strongly dependent on the reactive gas content in the gas mixture. Many works are focused on the ZnO properties dependence on oxygen content [19,20,22,26]. In the work of [19], it was reported that the higher thicknesses are obtained with the lowest presence of oxygen in the gas mixture. Since our main goal was to deposit thick ZnO coatings, we were focused on maximising the deposition rate by using minimum $O_{2}$, however, achieving the right stoichiometry of the ZnO film, which was proved by XRD results. Another study [26] also shows that (002) peak position shifts depending on the $O_{2}$ content and corresponds to the bulk ZnO when the oxygen content is around 50%, which correlates with the results reported in this work.

In a large number of works [19,20,21,22,23,24,25,26,27], ZnO was sputtered by using a planar Zn cathode, sometimes comparing it to sputtering from the ZnO cathode material. There were no examples of inverted or cylindrical source utilisation found in the literature. However, in the case of $TiO_{2}$ sputtering, one work reports a similar inverted magnetron approach [14]. Duarte et al., in their work, use an IM source with permanent magnets and a sample holder that can change its position inside the hollow cathode. They show the changes in the thickness of the deposits depending on oxygen content and sample holder position. As in the case of ZnO, the thicknesses for $TiO_{2}$ were no higher than 1000 nm, which makes it hard to compare with our study.

It is important to note that in the standard reactive MS, the poisoning (the compound growing through the cathode material) of the cathode might be a problem, which can significantly reduce the deposition rate. In many cases, a cathode cleaning process may be necessary to ensure controllable starting conditions. The cathode cleaning is a standard procedure that, in the case of a planar magnetron, can be realised by using a shutter, which separates the cathode from the substrate. To clean a cathode, a short sputtering run must be performed in the inert gas atmosphere [22]. In our case, probably due to the positive power connection, the ZnO creation occurred on the sample holder, and no sufficient cathode poisoning was noted.

The initial idea of the target manufacturing by IM was the utilisation of the vertical sample holder, which can host several targets simultaneously. However, the high sensitivity to the symmetry of the sample holder (due to the positive charge connection) does not allow us to obtain repeatable results. Moreover, the deposition rate was two to five times lower in comparison to the horizontally placed sample holder.

### 4.2. Material-Saving Approach

In this study, it was not possible to calculate the data related to the material utilisation since the prototype is not yet optimised in terms of its dimensions. However, in comparison to the previous study [7], no visible loss in the vacuum chamber or outside of the closed working environment has been observed. The cylindrical cathode acted as a self-shield, however, the deposition of the compound (such as ZnO) could lead to additional losses in the form of powder or flakes that may accumulate on the inactive parts of the cathode. Proper magnetic field configuration can prevent this problem. The shape of the sample holder is also a challenge since it will collect the material, which must be then recovered. Therefore, one of the tasks is to have a higher possible effective surface of the sample holder.

The preparation of the cathode for any MS configuration demands a significant material volume, unlike other techniques used for target manufacturing [9,10]. Therefore, we suggest MS as a potential competitor with other techniques in case of producing multiple targets in a single deposition run, which would minimise the relative material losses. However, it is crucial to validate the ability to deposit a uniform thickness on multiple targets in a single sputtering run in the IM configuration.

### 4.3. Targets Manufacturing

The maximum deposition rate achieved in this work was on the level of 200 nm/min, which is almost 10 times higher than reported in the work [23], where the authors address the importance of a high sputtering rate. It should be noted that in many studies [23,24,25], ZnO is deposited as a transparent conductive oxide with a thickness ranging from 50 to 2000 nm, and its electrical and optical properties are emphasised over higher deposition rates.

In our case, the goal of the ZnO target is the maximisation of radionuclide yield. Thus, the thickness should be uniform to exploit the best energy range with a higher probability of producing the radionuclide of interest, and the irradiation should be performed at the maximum sustainable proton current, as higher proton beam currents lead to higher radionuclide yields. This means that the adhesion of the target material to the backing must ensure good thermal exchange during the irradiation to keep the target intact before the radiochemistry dissolution step. This result can be achieved with the sputtering deposition [36].

Regarding the density, despite the bulk material being preferable for maximising radionuclidic yield, it could make the future dissolution step more complicated and slower. Thus, the density on the level 90–95% will be a good compromise between the requirements [29]. In our case, the relative density reached 70% of the bulk, probably due to the presence of the flakes and droplets. Nevertheless, the integrity of our targets after the irradiation tests were confirmed by visual inspection. However, the presence of defects in the deposition should be minimised to avoid detrimental effects (detachment or delamination) during the irradiation.

As can be noted from Figure 4b, the film exhibits growing defects right from the initial stages. These defects, commonly known as nodular defects, are discussed in [37] and typically form on seeds of various origins. In our case, it could be a combination of various factors, such as the occurrence of arcs and flakes, which are associated with the sputtering process and not with the substrate nature or preparation. One of the possible flake sources can be a flaking of redeposited nodules from the cathode surface. Reactive MS processes with a planar cathode can lead to this phenomenon, where some parts of the cathode’s area, such as the centre and edge, have a low plasma density and do not undergo sputtering. Due to the internal stresses, the redeposited material grows in the form of filaments. The authors in [37] indicate that such phenomena can be completely avoided in an inverted magnetron. In our case, this problem persisted due to the incomplete erosion of the cathode surface. To avoid these inhomogeneities and film defects, the sample holder can be placed in the top position with respect to the cathode surface. This also can contribute to higher process stability.

## 5. Conclusions

This work demonstrates the feasibility of an inverted magnetron prototype dedicated to solid target manufacturing aimed at radioisotope production for nuclear medicine applications. The material-saving approach described in this work can be exploited for different applications where highly expensive materials are used.

The first two ${}^{nat}ZnO$ onto *Nb* targets were obtained, which can be a starting material for the interesting theranostic ${}^{67}Cu$ radionuclide production. The study demonstrates the ability to grow an oxide film of 20–30 μm within a two hours time frame. The XRD analysis confirms the absence of Zn contaminants in the film and shows a preferable (002) crystal growth. The deposition rate reported in this work reaches 200 nm/min.

Additionally, the targets exhibited acceptable thermo-mechanical properties during short irradiation beam runs. Although the irradiation tests were successful, it is important to note that the real target thickness must be at least 500 μm for the proper isotope production [17]. Therefore, the achieved results are the first step of the further investigation of the MS technique and, in particular, the IM source.

## Figures and Tables

**Figure 1 materials-16-03810-f001:**
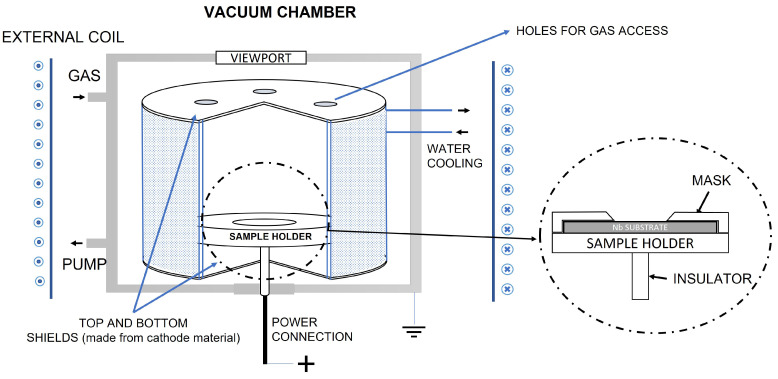
Scheme of the Inverted Magnetron (IM) sputtering system with the fixed sample holder. The sample holder can host a single Ø24 mm *Nb* coin to manufacture a single solid target with the Ø10 mm deposition area.

**Figure 2 materials-16-03810-f002:**
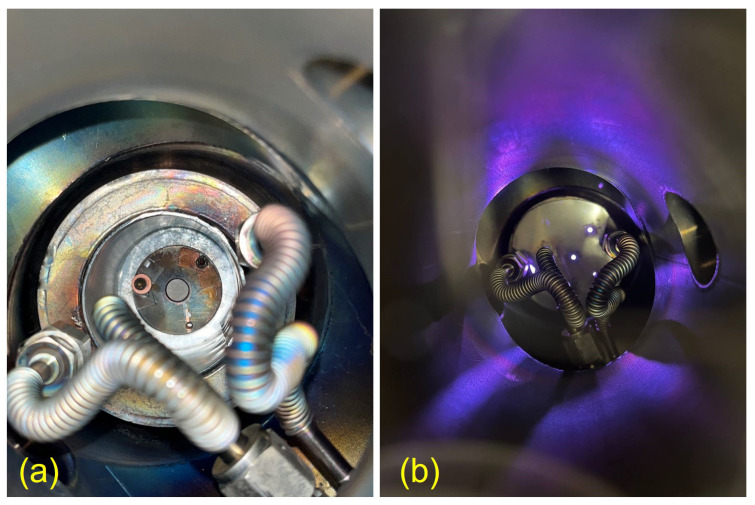
IM source in the vacuum sputtering chamber: (**a**) before the deposition with the mounted sample holder and (**b**) during the deposition with the top shield assembled.

**Figure 3 materials-16-03810-f003:**
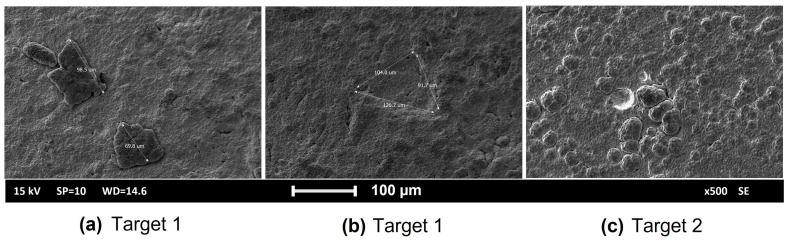
Defects on the surface of produced solid targets: (**a**,**b**)—Target 1; (**c**)—Target 2.

**Figure 4 materials-16-03810-f004:**
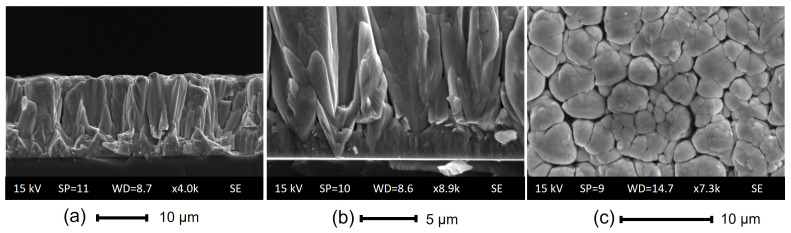
SEM analysis of ZnO deposition onto Si substrate (exp 8): (**a**,**b**) cross-section and (**c**) morphology.

**Figure 5 materials-16-03810-f005:**
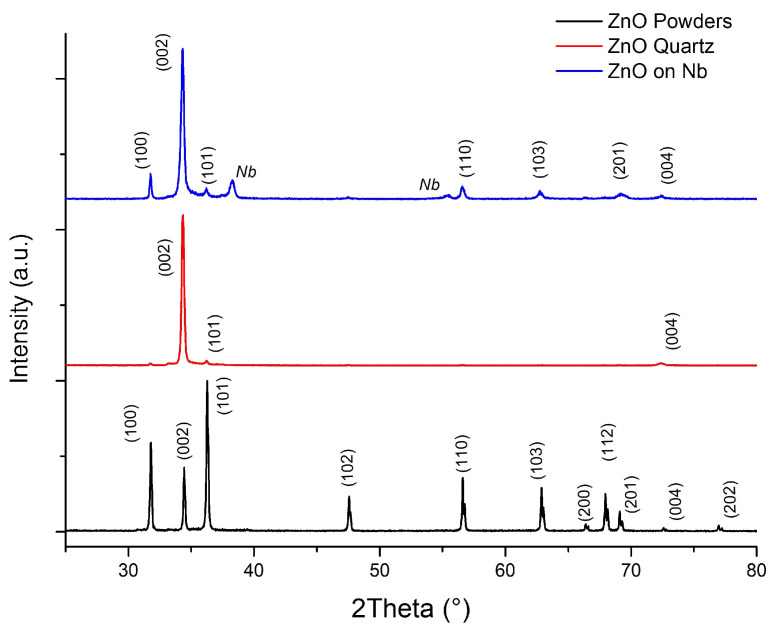
Diffractograms of ZnO films sputtered onto Nb and quartz substrates with respect to the diffractogram of ZnO powder.

**Figure 6 materials-16-03810-f006:**
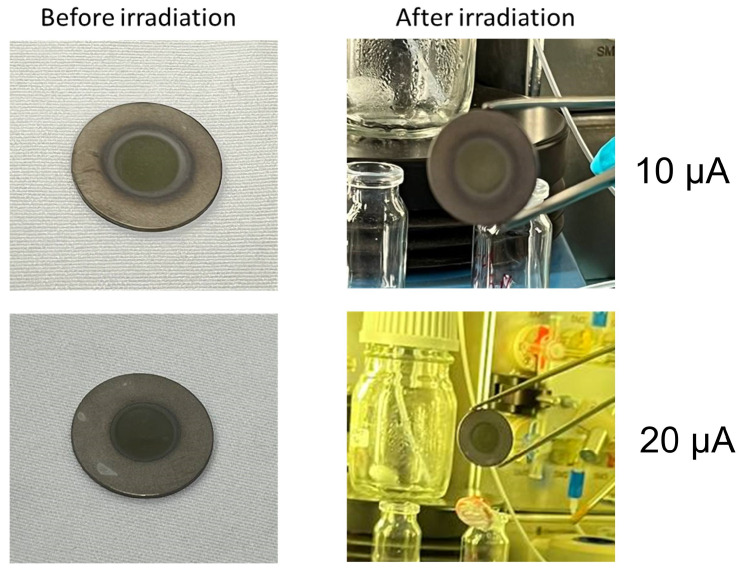
*ZnO* sputtered solid targets: before (**left**) and after (**right**) the irradiations under the 19 MeV proton beam of the TR19 ASCI cyclotron (Negrar, Verona, Italy).

**Table 1 materials-16-03810-t001:** Deposition parameters and results of *ZnO* films deposited with a constant current of 0.7 A.

Experiment Number	Sputtering Voltage [V]	Time [min]	Thickness [μm]	Δ M [ mg]	Density * [%]	Substrate
Exp 8	460	115	15	-	-	Si/Quartz
Target 1	500	130	20	6.34	4.1	Nb coin
Target 2	550	150	30	9.17	3.9	Nb coin

* Density is estimated by the weight and thickness measurements.

## Data Availability

No new data were created or analysed in this study. Data sharing is not applicable to this article.

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
