# Peer review of "First Results on Zinc Oxide Thick Film Deposition by Inverted Magnetron Sputtering for Cyclotron Solid Targets Production"

_materials, 2023, doi:10.3390/ma16103810_

Round 1
Reviewer 1 Report
The manuscript concerns deposition of thick ZnO coatings on Nb supports using novel inverted magnetron sputtering technique. Thus, its novelty includes new deposition technique applied to reactive sputtering of thick oxide coatings and the work deserves publications. However, number of improvements can be suggested:
1. the terminology used in the work is very confusing for the people working in conventional sputtering field: cathode is usually a target there and the sputtered particles are deposited on the substrate as a coating. In the current work, cathode material is not called the target (despite being a source of metal for the deposition) and instead of it, the deposited film is called "target" (it might be OK for cyclotron irradiation but not for deposition) and Nb substrate is the highly charged anode whereas the conventional anode is missing. Assuming that the manuscript aims at the conventional sputtering community, the terminology should be modified according to this field.
2. The information concerning deposition and use of the radionuclides in the introduction is rather remotely related to the content of the work: deposition of ZnO. It is recommended that the authors focus more on the deposition of such oxides, they should also avoid long paragraph on the projects supporting this work (move them into Acknowledgements) and only briefly mention potential applications of ZnO films as a host for various radionuclides for theranostic applications.
3. An inverted magnetron principles should be described in better way. Substrate position was not clear for me from Fig. 1, I had to read throughout the paper to realize where it is. Cathode cylinder seems to be tight-sealed in Fig. 1, how the sputtering and reactive gas gets into it? Why horizontal positioning of the sample holder was better that vertical? Was sample holder fixed or rotated? What is the role of the mask...was it also under tension or it was isolated from the substrate holder?
4. reactive sputtering was discussed only very briefly just saying that cathode (=target) poisoning did not occur. However, this seems to be an oversimplification of the story because of the formation of stoichiometric ZnO (this part of an experimental work is missing and can be assumed only based on X-ray diffractograms) on the substrate, heavy activation of Zn cathode (target) by Ar bombardment and high affinity of oxygen to all metals. One can also expect a strong influence of the amount of oxygen on the structure, stoichiometry and resulting properties, which was also missing.
5. The presence of droplets and large flakes in the coatings suggest that the deposition conditions involved arcs producing them. The authors should specify their role in the practical applications. - are these defects detrimental or not important at all (they should be definitely avoided in engineering applications)?
6. The results of the irradiation tests are extremely brief - "nothing visible". Is there a possibility for more scientific approach in the characterization of possible changes in the coating during such tests? What is their relevance to the real applications when radionuclide would be present in the coating?
6. The Results part contains many considerations which should be involved in the Discussion and the Discussion is then rather limited. More structured rearrangements between results and Discussion is therefore recommended.
The above remarks are mostly formal and can be solved within the category "major changes". After they are solved, the work can be recommended for publication.
Few sentences were difficult to understand:
e.g. p. 2, row 39:... may sustain a high beam power hitting on it and...
p.2, r. 50 ...does not allow to make a step to use an isotopic-enriched...material.
p. 2 , r. 62-63... High-density plasma... this sentence does not stick to the rest of this paragraph.
Reviewer 2 Report
The authors write: "The sputtered targets showed no visible signs of degradation or change after irradiation testing." Visible signs are not enough!!!
The energy of protons is quite high (19 MeV) and it is well known that already electrons with energies > 300 keV create point defects. See review paper:
https://doi.org/10.1016/j.nimb.2010.05.053
and cited papers [49, 8] therein.
The work has not been carried out accordingly and gives incorrect data on the radiation resistance of ZnO
Reviewer 3 Report
In this work, an inverted magnetron prototype for the deposition of tens of μm film onto different substrates was developed. The deposited materials were characterized. The work is original and interesting. I recommend the manuscript to be published after revision.
I. In Table 1, it shows the Exp 8-10 results. The reviewer understood that 'Exp 8-10' was the author's nomenclature for the experimental data. However, this will make the reader feel like it is a technical report. The reviewer suggests that the authors describe them in detail not just name as 'Exp 8-10' (possibly as supporting material).
II. The scale bars for all images are small. They should be re-drawn.
III. In Discussion, many arguments are not supported by literature. The authors should compare the technique in this manuscript with similar techniques in published papers to highlight the advancement of this technology.
IV. The conclusions are simplistic and can be expanded.
Reviewer 4 Report
Referee’s Report on Ms: materials-2357684
Title: „First Results on Zinc Oxide Thick Film Deposition by Inverted Magnetron Sputtering for Cyclotron Solid Targets Production”
by Alisa Kotliarenko, Oscar Azzolini, Sara Cisternino, et al.
Considering the growing interest in 67Cu in theranostics and the availability of compact cyclotrons, the demand for preparing thick enriched 68Zn or 70Zn targets for 67Cu production has intensified. In this sense, the topic of this manuscript is important and the results are of interest for diagnostic nuclear medicine. The authors have developed a prototype sputtering system using an inverted magnetron source and have demonstrated that it is feasible to deposit considerably thick ZnO layers (20-30 mm) with good thermo-mechanical properties. The article is well written and easy to read. The results and discussion are supported by relevant references.
My opinion is that the present manuscript can be accepted for publication after a minor revision.
I have some remarks and comments which should be considered by the authors:
- The determination of density defined as weight per unit volume is not quite accurate and the density of 73%-71% of bulk ZnO is not a specified value. It is better to give the numerically calculated density values than those in percentages. From which paper did the authors take the bulk ZnO density and for what kind of ZnO material? The corresponding citation should be added. Judging by the percentage densities of 71-73%, the layers are quite porous, which would affect the actual density of the layers.
- Isn't the high porosity a disadvantage in further application of these ZnO layers? What is the authors’ consideration about this?
- According to the XRD spectra, the deposited ZnO films have a wurtzite polycrystalline structure with a preferential orientation (002), i.e. crystallographic c-axis oriented. It would be worth to calculate the nanocrystallites size from this single XRD peak. Apparently, such a polycrystalline structure contributes to the observed thermo-mechanical stability of ZnO layers under irradiation.
- Since “the first natZnO targets” have been tested “under the proton beam of a medical cyclotron”, this experiment should be investigated more extensively and its results should be discussed in more detail. How, from just a visual inspection of the irradiated ZnO coins, can it be said that they can be further used to produce 67Cu, which is, as far as I understood, the main purpose of the authors?
- In order to obtain 67Cu, significantly thicker layers (several hundreds mm) are required, which at the given configuration and forms of the IMS prototype is limited by the deposition time due to “the flakes formation, which resulted in the arcs and instability of the process.” How do the authors intend to overcome this problem?
- In Fig. 3a, 3c, the denotations are barely visible, the letters and numbers should be enlarged, and the corresponding lines should be thicker.

Minor editing of English language required
Round 2
Reviewer 2 Report
After reasonable revision, this manuscript can be recommended for publication.
Author Response
Thank you very much for your kind assistance.
Reviewer 3 Report
It can be accepted.
Author Response
We would like to express our sincere gratitude to the reviewer for their thorough evaluation and valuable feedback on our manuscript.